# A causal role for the precuneus in network-wide theta and gamma oscillatory activity during complex memory retrieval

Melissa Hebscher[1,2,3]*, Jed A Meltzer[1,2], Asaf Gilboa[1]*

[1]Rotman Research Institute, Baycrest, Toronto, Canada; [2]Department of Psychology, University of Toronto, Toronto, Canada; [3]Department of Medical Social Sciences, Northwestern University Feinberg School of Medicine, Chicago, United States

**Abstract** Complex memory of personal events is thought to depend on coordinated reinstatement of cortical representations by the medial temporal lobes (MTL). MTL-cortical theta and gamma coupling is believed to mediate such coordination, but which cortical structures are critical for retrieval and how they influence oscillatory coupling is unclear. We used magnetoencephalography (MEG) combined with continuous theta burst stimulation (cTBS) to (i) clarify the roles of theta and gamma oscillations in network-wide communication during naturalistic memory retrieval, and (ii) understand the causal relationship between cortical network nodes and oscillatory communication. Retrieval was associated with MTL-posterior neocortical theta phase coupling and theta-gamma phase-amplitude coupling relative to a rest period. Precuneus cTBS altered MTL-neocortical communication by modulating theta and gamma oscillatory coupling. These findings provide a mechanistic account for MTL-cortical communication and demonstrate that the precuneus is a critical cortical node of oscillatory activity, coordinating cross-regional interactions that drive remembering.
DOI: https://doi.org/10.7554/eLife.43114.001

*For correspondence:
melissa.hebscher@northwestern.edu (MH);
agilboa@research.baycrest.org (AG)

**Competing interests:** The authors declare that no competing interests exist.

## Introduction

Detailed, complex memories of personal events can last a lifetime and be brought to mind at will. Current models suggest that this ability depends on the hippocampus and surrounding MTL regions, as well as the coordinated reinstatement of retrieved information from neocortical regions (*McClelland et al., 1995*; *Rolls, 2000*). The critical role of the MTL in retrieval of detailed personal memories has been extensively demonstrated (*Nadel and Moscovitch, 1997*; *Rosenbaum et al., 2008*; *Scoville and Milner, 1957*). However, there is little empirical evidence indicating which neo-cortical structures are crucially involved in the reinstatement of detailed personal memories. Causal evidence is also lacking for a mechanistic account of how such coordinated reactivation occurs during complex personal memory retrieval.

Interactions between MTL and medial parietal regions are thought to be particularly important for representing the spatial context of an event, a central contributor to the vivid recollection of memories (*Burgess et al., 2001a*; *Burgess et al., 2001b*; *Hassabis and Maguire, 2009*; *Robin et al., 2015*). While previous studies have shown that MTL and medial parietal structures are associated with spatial aspects of personal memory retrieval (*Freton et al., 2014*; *Hebscher et al., 2018*; *St Jacques et al., 2017*), they are unable to determine whether these regions are critical for retrieval, or how they may communicate with one another to allow for information transfer.

**eLife digest** When you recall an event from your past, such as a meal you ate last week, many regions of your brain become active. The coordinated activity of these regions enables you to recall the event in detail. This coordination depends on rhythmic waves of electrical activity called neural oscillations. These arise whenever large numbers of neurons synchronize when they fire. Electrodes on the scalp can be used to measure neural oscillations. Recordings show that the number of times each wave repeats per second (also known as the frequency of the oscillation), varies from one brain region to the next.

Two types of oscillations are particularly important for memory: theta waves and gamma waves. Theta waves repeat between three and seven times every second, and help coordinate activity between areas of the brain that are far apart. Gamma waves are faster, repeating 65 to 85 times per second, and help to support activity within individual regions of the brain. Importantly, theta and gamma waves also interact.

Hebscher et al. set out to understand whether interactions between theta and gamma waves help us to recall personal memories. Healthy volunteers were asked to recall memories in response to cues such as 'my kitchen', while sitting inside a brain scanner. As predicted, interactions between theta and gamma waves contributed to memory recall. Theta waves recorded from the medial temporal lobe, a region deep within the brain, altered gamma waves in another area called the precuneus. The latter forms part of the inner surface of the brain where the two hemispheres face one another, and is important for memory vividness and visual imagery. Hebscher et al. briefly disrupted the activity of the precuneus by applying harmless magnetic fields to the scalp above it. Doing so altered theta-gamma interactions across the brain, which was related to reduced vividness of the memories.

Remembering events from our past is fundamental to our sense of self and our interactions with others. The results presented by Hebscher et al. show that reducing the activity of a single brain region, the precuneus, impairs memory recall. It does so by disrupting interactions between oscillations throughout the brain. This raises the possibility that stimulating the brain to enhance – rather than disrupt – oscillations could have the opposite effect and improve memory. Future studies could investigate whether enhancing oscillations could help to treat memory disorders.

DOI: https://doi.org/10.7554/eLife.43114.002

Neural oscillations may be a mechanism by which such widespread regions communicate during memory retrieval. Theta (3–7 Hz) oscillations are hypothesized to mediate MTL-neocortical orchestration during memory retrieval (*Battaglia et al., 2011*), and theta phase coherence between regions has been identified in working (*Payne and Kounios, 2009*), spatial (*Kaplan et al., 2014*; *Kaplan et al., 2017*), and autobiographical memory (*Eckart et al., 2014*; *Foster et al., 2013*). Theta phase may also modulate spatially distributed local gamma oscillations through phase-amplitude coupling (PAC), a form of cross-frequency coupling (*Sirota et al., 2008*). Theta-gamma PAC is thought to be a mechanism for communication between distributed regions during cognitive processes (*Canolty et al., 2007*) and has been identified in the human MTL (*Axmacher et al., 2010*; *Staudigl and Hanslmayr, 2013*) and neocortex during memory (*Canolty et al., 2007*; *Kaplan et al., 2014*; *Sauseng et al., 2009*; *van der Meij et al., 2012*). In animals, theta oscillations reflect the organization of complex spatial memories as they unfold and are critical for accurate reinstatement of conceptually meaningful representations (*Wikenheiser and Redish, 2015*). Disruption of this oscillatory activity interferes with the integrated representation of complex, temporally extended memories (*Colgin, 2016*).

Human studies of complex naturalistic memories have thus far only demonstrated the existence of theta phase synchronization as a correlate of memory retrieval (*Foster et al., 2013*; *Fuentemilla et al., 2014*). Theta-gamma coupling has only been demonstrated in relatively simple lab-based experiments (*Axmacher et al., 2010*; *Canolty et al., 2007*; *Kaplan et al., 2014*; *Sauseng et al., 2009*) and no study has demonstrated its critical contribution to complex mnemonic functions. Importantly, memory for items presented during lab-based experiments recruits different neural substrates than memory for real-life events, which is considered to be more contextually rich,

self-focused, and complex (*Chen et al., 2017*; *Gilboa, 2004*; *McDermott et al., 2009*). Here, we aim to (i) clarify the roles of oscillations within and across network nodes during retrieval of detailed complex personal memories and (ii) understand the causal relationship between these network nodes and oscillatory communication. The present study elucidates the communication between regions involved in autobiographical memory (AM) by measuring theta phase coupling and theta-gamma PAC during memory retrieval. Participants performed an AM task in which they were cued with familiar words and rated the subjective quality of memories (*Figure 1*).

We further use continuous theta burst stimulation (cTBS) to examine whether the precuneus is causally involved in oscillatory communication between regions during complex memory retrieval. Previous studies have used parietal neurostimulation to alter autobiographical (*Bonnici et al., 2018*; *Thakral et al., 2017*) and episodic memory (*Bonnì et al., 2015*; *Nilakantan et al., 2017*; *Wang et al., 2014*; *Wang and Voss, 2015*; *Yazar et al., 2014*), some of which have shown associated and sustained alterations of neural activity (*Nilakantan et al., 2017*; *Wang et al., 2014*; *Wang and Voss, 2015*). In the present study, we used cTBS to directly suppress neural activity in the precuneus in order to observe system-level changes in activity, specifically in the MTL and other structures that comprise the autobiographical memory network. The precuneus is a highly structurally and functionally connected association area (*Cavanna and Trimble, 2006*) that demonstrates theta phase coupling during AM (*Fuentemilla et al., 2014*), and has a causal behavioural role in episodic memory (*Bonnì et al., 2015*; *Koch et al., 2018*). We therefore predicted that precuneus stimulation would affect neural activity both within the precuneus and in regions functionally connected to the precuneus. Behaviorally, we predicted that stimulation would lead to differences in subjective aspects of complex memory retrieval. Specifically, we predicted that stimulation would alter the perspective from which memories were recollected based on the precuneus' established role in spatial perspective representations during memory (*Freton et al., 2014*; *Hebscher et al., 2018*; *St Jacques et al., 2017*).

## Results

Twenty-three healthy young participants (14 females, mean age = 26.3, range = 19–36) were tested on a within-subjects combined transcranial magnetic stimulation (TMS) and MEG paradigm. Participants received continuous theta burst stimulation (cTBS) to their left precuneus and to a control region (vertex) on separate days at least 24 hr apart (mean = 5.4 days, SD = 5.3 days). Immediately following cTBS, participants completed an AM task inside the MEG scanner which was located nearby. In each session participants recalled memories cued by familiar items and rated each memory on scales measuring subjective AM (*Figure 1*).

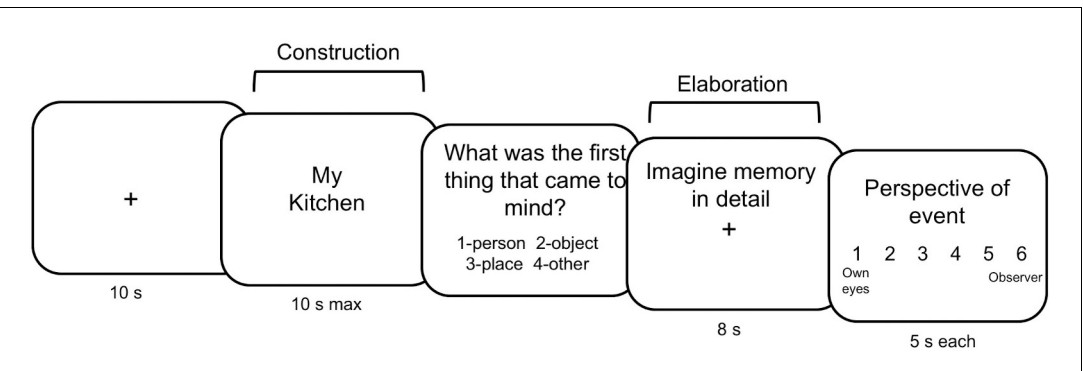

**Figure 1.** Autobiographical memory paradigm. Participants were cued with familiar words (locations, people, objects), and told to recall a past event in relation to this word. Construction was terminated when participants indicated they had an event in mind via button press.
DOI: https://doi.org/10.7554/eLife.43114.003

## Behavioral results

We first investigated whether precuneus stimulation would lead to differences in the quality of memory retrieval by comparing rating scales from the AM task between precuneus and vertex stimulation sessions. Rating scales included memory vividness, ease of recall, and perspective rating (first- versus third-person perspective). We performed a series of analyses of variance (ANOVAs) using rating scales for precuneus and vertex sessions as within-subjects factors and session order (i.e. counterbalancing order: precuneus stimulation or vertex stimulation first) as the between-subjects factor, to account for potential effects of counterbalancing order on subjective memory. These analyses revealed a significant interaction between session order and session type on vividness ratings (F $(1,21)$ = 5.10, p=0.035, $\eta p^2$ = 0.195) but not on ease of recall (F$(1,21)$ = 0.84, p=0.369, $\eta p^2$ = 0.039) or perspective ratings (F$(1,21)$ = 0.10, p=0.755, $\eta p^2$ = 0.005). There was a main effect of stimulation type on vividness ratings when accounting for counterbalancing order (F$(1,21)$ = 4.83, p=0.039, $\eta p^2$ = 0.187), such that precuneus stimulation led to lower vividness ratings. There was no significant effect of stimulation on effort or perspective ratings (all p's > 0.28). These findings suggest that the order of counterbalancing significantly interacted with the effects of stimulation on memory vividness.

Based on the significant interaction between session order and session type, we performed a post-hoc exploratory analysis on vividness ratings using each participant's first session only, such that we compared participants with precuneus stimulation first (n = 12) to those with vertex stimulation first (n = 11). This analysis had a between-subjects design, which several previous parietal-TMS studies have used (cf. *Yazar et al., 2014*; *Bonnì et al., 2015*; *Wang and Voss, 2015*). Independent-samples t-tests revealed a significant difference in vividness ratings, such that precuneus stimulation led to less vivid memories compared to vertex stimulation (t$(21)$= −2.34, p=0.030, CI [−0.98 −0.06], d = −0.97) (see *Figure 2A*). Although there was no significant interaction between session order and session type on effort rating, we also found a significant difference between stimulation sessions on ease of recall ratings, such that precuneus stimulation led to more effortful recall compared to vertex stimulation (t$(21)$ = −2.62, p=0.016, CI [−0.98 −0.11], d = −1.09) (*Figure 2B*). Comparison of each participant's second session revealed no significant differences between stimulation sessions on vividness (t$(21)$= 1.55, p=0.136, CI [−0.16 1.1], d = 0.65) or ease of recall (t$(21)$= 1.10, p=0.287, CI [−0.28 91], d = 0.46) (*Figure 2—figure supplement 1*).

## MEG correlates of complex personal memory

To gain a better understanding of the oscillatory underpinnings of complex memory retrieval, we examined MEG data at vertex stimulation sessions. The subsequent sections describe results for theta power, phase coupling, and theta-gamma phase-amplitude coupling.

### Theta power during memory retrieval

We first examined source-estimated maps of changes in theta power (3–7 Hz) for memory retrieval at vertex stimulation sessions. Source estimated maps of theta power change were produced by comparing early memory elaboration, when participants were retrieving the details and recollecting the memory, to an inter-trial rest period of equal length. Voxel-wise t-tests revealed that compared to rest, memory elaboration was associated with theta power increases bilaterally in the occipital lobe, precuneus, inferior and superior parietal lobes including the angular gyrus, retrosplenial cortex, MTL, medial prefrontal cortex (mPFC), and cerebellum (*Figure 3A*; *Table 1A*). Theta power maps largely corresponded with regions commonly activated in autobiographical memory studies (*Svoboda et al., 2006*), suggesting that theta oscillations may underlie widespread neural activation during AM. Theta power increase maps were subsequently used to determine seed regions for phase and phase-amplitude coupling analyses.

### Phase coupling during memory retrieval

To examine theta phase coupling during memory retrieval, we computed Weighted Phase Lag Index (wPLI) of source-estimated time series at vertex stimulation sessions using a right MTL seed. Participant-specific seeds were selected as the virtual voxel (10 mm$^3$) showing the greatest theta power increase during memory retrieval relative to rest within the right MTL (anatomically defined). Lateralization of the MTL seed was determined by the location of maximum theta power increase during

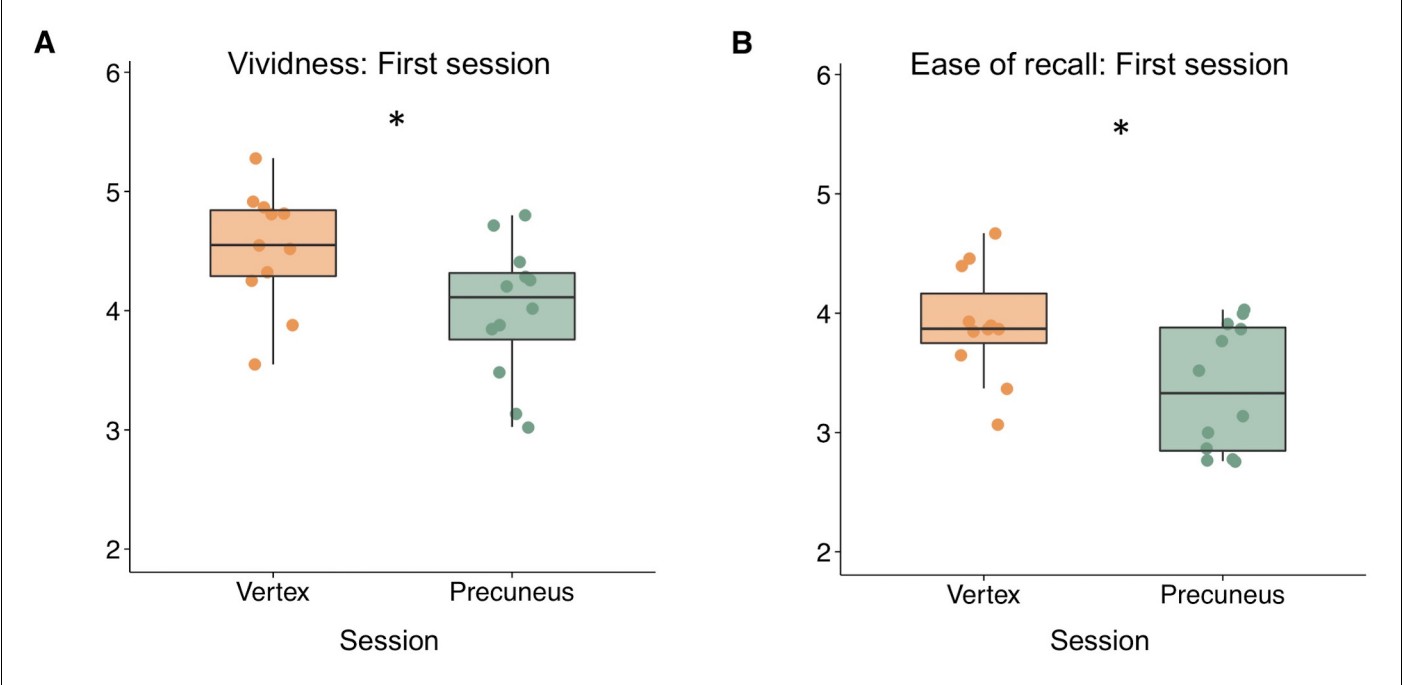

**Figure 2.** Exploratory between-subjects analysis showing that precuneus stimulation leads to (A) reduced vividness ratings and (B) reduced ease of recall compared to vertex stimulation when considering only the first session for each participant. Asterisks indicate a significant difference between precuneus and vertex stimulation (p<0.05).

DOI: https://doi.org/10.7554/eLife.43114.004

The following source data and figure supplement are available for figure 2:

**Source data 1.** Behavioural data.

DOI: https://doi.org/10.7554/eLife.43114.006

**Figure supplement 1.** Exploratory between-subjects analysis showing non-significant differences between precuneus and vertex stimulation for (A) vividness and (B) ease of recall, when considering only the second session for each participant.

DOI: https://doi.org/10.7554/eLife.43114.005

memory retrieval relative to rest, at the group level. Paired-sample t-tests comparing wPLI during early memory elaboration to rest revealed increased theta phase coupling between the right MTL seed and a cluster in the right occipital lobe including the fusiform gyrus, secondary visual cortex, and associative visual cortex (*Figure 3B*, *Table 1B*), although note that exact locations are difficult to determine using MEG source estimation procedures. These findings show that MTL and posterior occipital regions communicate via theta phase coupling during AM retrieval.

To determine if MTL-posterior cortical phase coupling is associated with subjective memory measures we exported average wPLI values for the significant occipital lobe cluster for each participant. Theta phase coupling was not significantly correlated with vividness, effort, or perspective ratings (all p's > 0.395).

## PAC during memory retrieval

To determine whether MTL theta modulates neocortical gamma during autobiographical memory retrieval, we assessed theta-gamma phase amplitude coupling. We calculated phase locking value modulation index (PLV-MI) between theta phase of an MTL seed and gamma amplitude of left precuneus, right mPFC, and left temporoparietal junction (TPJ) regions of interest (ROI). In addition to the precuneus, mPFC and TPJ were chosen as ROIs as they are commonly activated cortical regions in AM studies (*Svoboda et al., 2006*), and showed increases in theta power during our task. Lateralization of the seed and ROIs were determined based on theta power maps, as described above (*Figure 3A*). Participant-specific ROIs were chosen as the voxels showing greatest theta power increase relative to rest within each anatomically defined region. Surrogate normalized simulation-

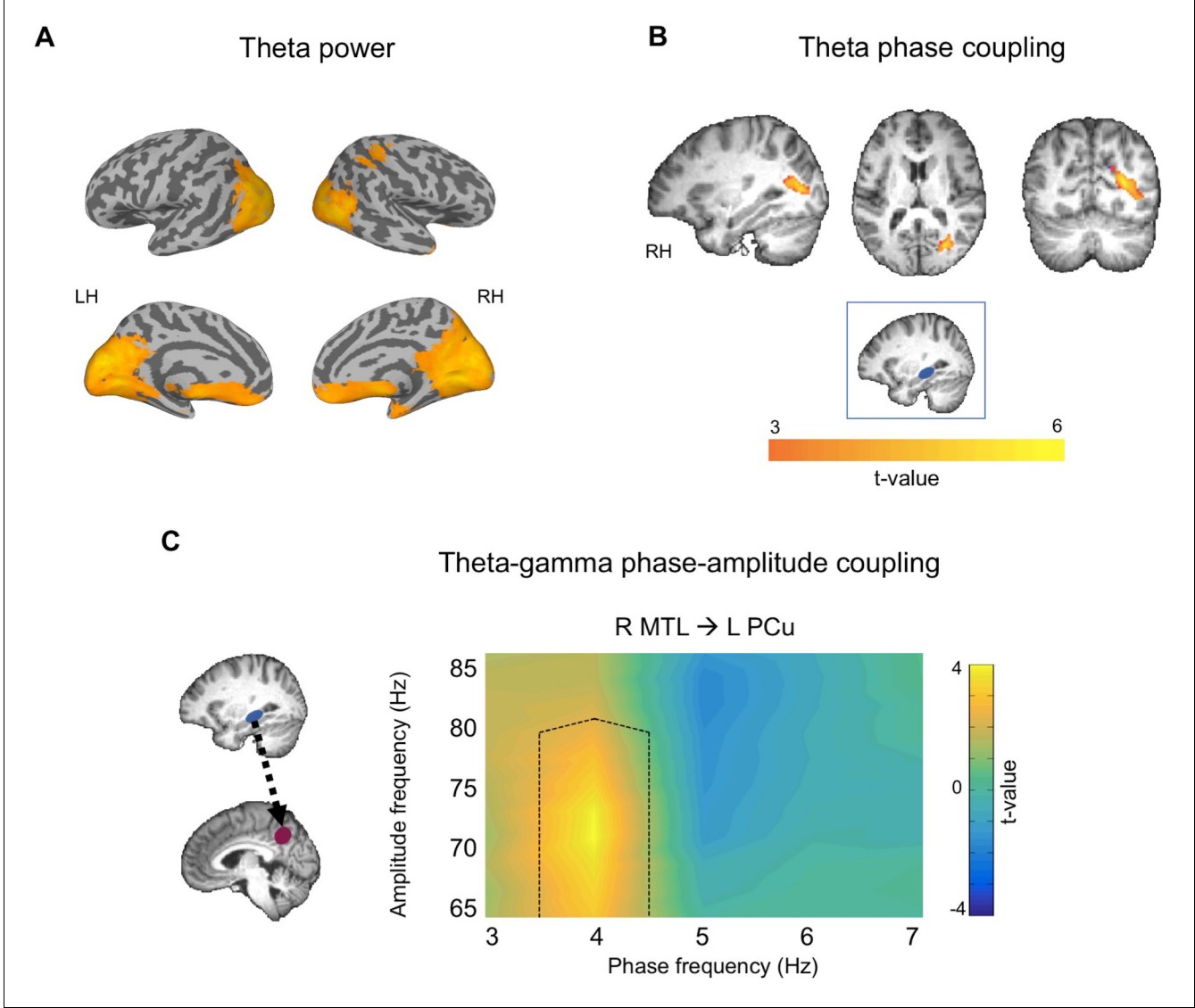

**Figure 3.** Theta activity during autobiographical memory retrieval for vertex stimulation sessions. (A) Theta power increases during AM retrieval relative to rest. (B) Increased theta phase coupling during AM retrieval using a right MTL seed (blue box). Theta power and phase coupling images displayed at p<0.005, cluster corrected. (C) Comodulogram showing theta-gamma phase-amplitude coupling between right MTL theta phase and left precuneus gamma amplitude. Black dotted lines on comodulogram shows areas of significantly different phase-amplitude coupling between memory and rest. Comodulogram displayed at p<0.05, cluster corrected.

DOI: https://doi.org/10.7554/eLife.43114.008

The following source data is available for figure 3:

**Source data 1.** Data used for plotting comodulograms.

DOI: https://doi.org/10.7554/eLife.43114.009

based cluster corrected t-tests revealed increased PAC between the left precuneus and right MTL during elaboration relative to rest (cluster p=0.025, t-sum = 30.22) (*Figure 3C*). There were no significant differences between PAC during memory and rest for right mPFC (cluster p=0.145, t-sum = 6.73) or left TPJ seeds (no clusters passed initial threshold).

We also tested for PAC between theta phase of each neocortical ROI and gamma amplitude of the MTL seed (the reverse direction as above), to determine the direction of theta phase modulation of gamma. This analysis revealed no significant differences in PAC during memory compared to rest for precuneus (no clusters passed initial threshold), mPFC (no clusters passed initial threshold), or

**Table 1.** Theta power and phase coupling cluster information.

| Label | Size (voxels) | T | X | Y | Z |
|---|---|---|---|---|---|
| **(A) Theta power: Elaboration > rest** | | | | | |
| Bilateral occipital lobe, precuneus, cingulate cortex, inferior parietal and superior parietal lobes, retrosplenial cortex, MTL, mPFC, cerebellum | 4141 | 7.38 | −27 | −70 | 29 |
| **(B) Theta phase coupling: Elaboration > rest** | | | | | |
| Right occipital lobe | 38 | 5.54 | 28 | −70 | 16 |
| **(C) Theta phase coupling: Precuneus > vertex** | | | | | |
| Left occipital lobe | 24 | −5.30 | -2 | −90 | 6 |

DOI: https://doi.org/10.7554/eLife.43114.007

TPJ seed regions (cluster p=0.234, t-sum = −2.27), demonstrating that MTL theta modulates precuneus gamma, but precuneus theta does not modulate MTL gamma. These results support previous claims that MTL theta plays a role in coordinating neocortical gamma, suggesting a means of information transfer between these widespread regions.

To determine if MTL-precuneus PAC is associated with subjective memory measures, we exported average PAC values for the previously identified significant cluster (*Figure 3C*) for each participant. This analysis revealed a significant positive correlation between MTL-precuneus PAC and vividness ratings (r = 0.483, p=0.020, CI [.06. 73]), but not ease of recall (r = 0.181, p=0.408, CI [−0.31. 59]) or perspective ratings (r = −0.050, p=0.820, CI [−0.36. 47]). These findings demonstrate that MTL-precuneus PAC is related to subjective memory vividness.

## Effects of precuneus stimulation on oscillatory activity
### Effects of precuneus stimulation on theta power
We tested the prediction that precuneus stimulation will affect neural oscillations by subjecting source-estimated maps for memory (relative to rest) to paired t-tests comparing theta power after precuneus vs. vertex stimulation. Results revealed no group differences between precuneus and vertex stimulation sessions during memory elaboration relative to rest for theta power (cluster p's > .005).

### Effects of precuneus stimulation on phase coupling
To test the prediction that precuneus stimulation will affect communication between regions, we compared theta phase coupling maps for memory elaboration relative to rest between precuneus and vertex stimulation sessions. A paired-sample t-test revealed that precuneus stimulation significantly reduced phase coupling between the right MTL seed and a cluster in the left occipital lobe, encroaching into the right occipital lobe (*Figure 4A*, *Table 1C*). Notably, this cluster did not overlap with the right occipital cluster demonstrating increased phase coupling during vertex stimulation sessions (*Figure 3A*). However, we observed a subthreshold effect of precuneus stimulation in the right occipital lobe (18 voxels at p<0.010), suggesting that precuneus stimulation may have decreased the phase coupling that was present at vertex stimulation sessions. These findings demonstrate that precuneus stimulation disrupts theta phase coupling between the right MTL and posterior neocortical regions, suggesting that the effects of stimulation are widespread and extend beyond the site of stimulation.

To determine if stimulation-induced changes in phase coupling are associated with stimulation-induced changes in subjective memory measures (vividness and effort ratings), we exported average wPLI values for the significant left occipital cluster identified previously (*Figure 4A*) for each participant. We calculated partial correlations between change in phase coupling and change in subjective memory, holding counterbalancing order constant. Change in theta phase coupling was not significantly correlated with change in vividness or effort rating (all p's > 0.346), suggesting that stimulation-induced changes in phase coupling and subjective memory measures are not related.

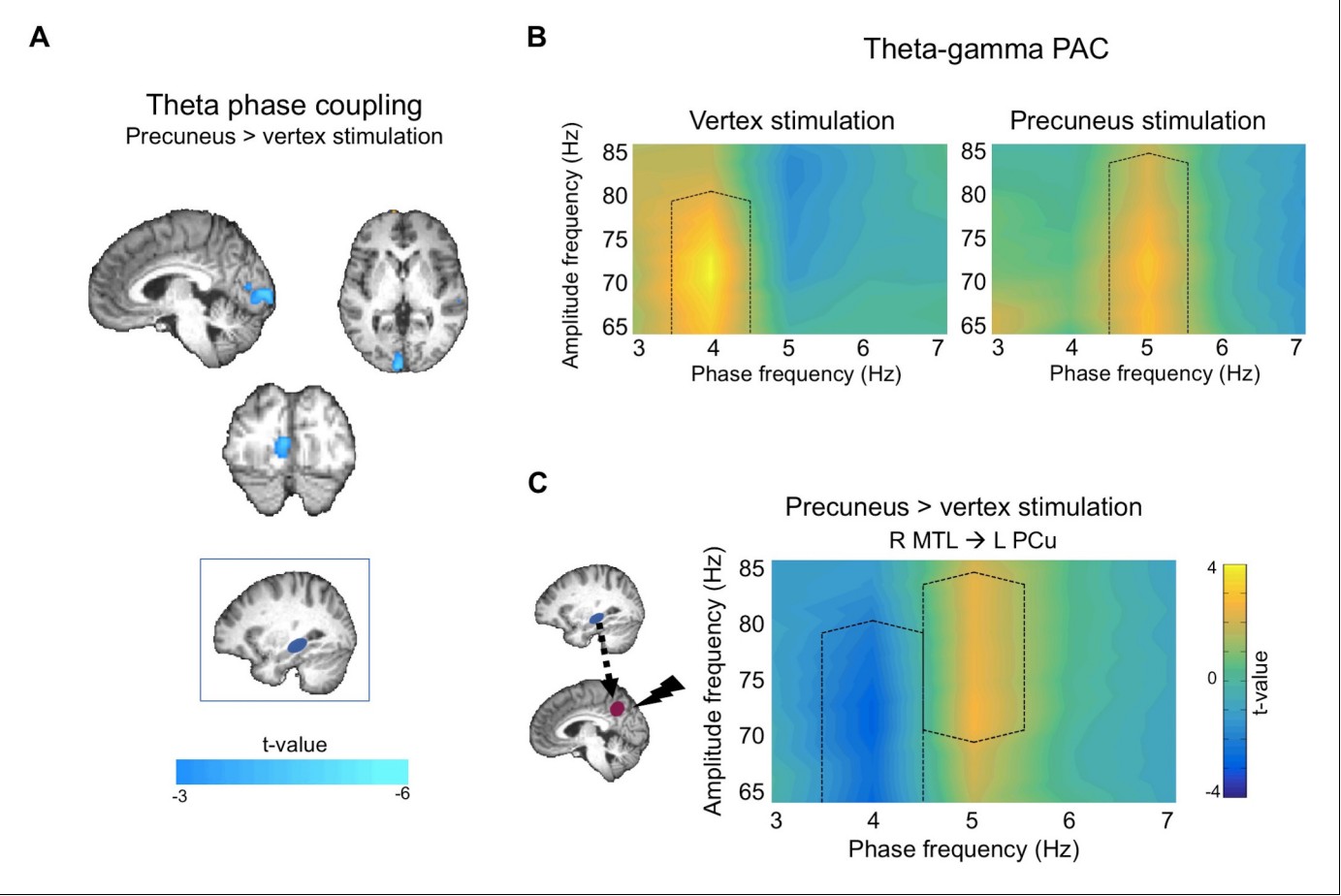

**Figure 4.** Effects of cTBS on theta activity during memory elaboration relative to rest. (**A**) Precuneus stimulation led to decreased theta phase coupling between a right MTL seed (blue box) and the left occipital lobe. Images displayed at p<0.005, cluster corrected. (**B**) Comodulograms showing theta-gamma phase-amplitude coupling separately for vertex (left) and precuneus (right) stimulation sessions, between right MTL theta phase and left precuneus gamma amplitude. Black dotted lines show areas of significantly different phase-amplitude coupling between memory and rest. (**C**) Comparison between comodulograms in (**B**), showing precuneus compared to vertex stimulation during memory retrieval, between right MTL theta phase and left precuneus gamma amplitude. Black dotted lines show areas of significantly different phase-amplitude coupling between precuneus and vertex stimulation. Comodulograms displayed at p<0.05, cluster corrected.

DOI: https://doi.org/10.7554/eLife.43114.010
The following source data and figure supplement are available for figure 4:

**Source data 1.** Data used for plotting comodulograms.
DOI: https://doi.org/10.7554/eLife.43114.012
**Figure supplement 1.** Phase-amplitude coupling using a broader amplitude frequency range reveals that effects are specific to high gamma.
DOI: https://doi.org/10.7554/eLife.43114.011

## Effects of precuneus stimulation on PAC

To further test the prediction that precuneus stimulation will affect network-wide oscillatory activity, PAC comodulograms for memory relative to rest following precuneus stimulation were compared to those following vertex stimulation sessions. Simulation-based cluster corrected t-tests revealed two significant clusters, demonstrating that precuneus stimulation altered right MTL- left precuneus PAC. Precuneus stimulation decreased coupling at 4 Hz (cluster p=0.024, t-sum = −20.51) and increased coupling at 5 Hz (cluster p=0.044, t-sum = 17.41) (*Figure 4C*). Notably, the decrease at 4 Hz corresponded with the increase at 4 Hz for vertex stimulation sessions (*Figure 3C*, *Figure 4B*, left), while the increase in 5 Hz corresponded with an increase at 5 Hz found for precuneus stimulation sessions (cluster p=0.011, t-sum = 30.61; *Figure 4B*, right). Precuneus stimulation did not significantly alter PAC between the right MTL and right mPFC (no clusters passed initial threshold) or left TPJ (cluster

p=0.115, t-sum = 13.84). Additional analyses measuring PAC between neocortical theta phase and MTL gamma amplitude revealed no significant differences between precuneus and vertex stimulation for any of the three ROIs (no clusters passed initial threshold), demonstrating that stimulation affects MTL theta phase modulation of precuneus gamma amplitude, and not the reverse. These findings thus show that precuneus plays a causal role in MTL-precuneus phase-amplitude coupling.

To determine if stimulation-induced changes in PAC are associated with stimulation-induced changes in subjective memory measures, we exported average PAC values for the two significant clusters identified above (*Figure 4C*) for each participant. We calculated partial correlations between change in PAC within each cluster separately, and change in subjective memory, holding counterbalancing order constant. Change in 4 Hz modulation of gamma was positively correlated with change in vividness, when accounting for counterbalancing order (partial r = 0.466, p=0.029, CI [.21. 68]), but not with change in effort rating (partial r = −0.322, p=0.143, CI [−0.71. 08]). Change in 5 Hz modulation of gamma was not significantly associated with change in vividness (partial r = 0.282, p=0.203, CI [−0.28. 66]) or effort (partial r = 0.364, p=0.095, CI [−0.30. 83]). Thus, greater stimulation-induced deficit in vividness is associated with greater reduction in 4 Hz phase modulation of gamma amplitude, regardless of counterbalancing order. These findings show that MTL-precuneus PAC is related to subjective memory, and suggest that precuneus is causally involved in this relationship.

## Additional analyses

### Control analysis: counterbalancing order

There was an interaction between counterbalancing order and stimulation type on subjective memory measures (*Figure 2*). We therefore also examined possible interactions between the effects of stimulation on coupling measures and counterbalancing order. We performed the same comparisons between precuneus and vertex stimulation sessions as described above, using counterbalancing order as a control variable. A paired-sample t-test on phase coupling maps revealed a similar cluster as in the original analysis (*Figure 4A*) in the left occipital lobe when holding counterbalancing order constant (30 voxels; t = −4.19; x = −2, y = −80, z = 10), and a second cluster in the right occipital lobe (25 voxels; t = −4.73; x = 17, y = −70, z = 25). Similarly, cluster corrected t-tests on PAC comodulograms using counterbalancing order as a control variable revealed the same pattern of results as described above (*Figure 4C*), including a decrease at 4 Hz (cluster p=0.022, t-sum = −20.51) and increase at 5 Hz (cluster p=0.029, t-sum = 19.48). These findings indicate that counterbalancing order did not influence stimulation's effects on phase coupling or PAC.

### Control analysis: gamma band specificity

The present study examined theta modulation of high-gamma, defined as 65–85 Hz (*Kaplan et al., 2014*), based on its reported consistency across a variety of studies (*Crone et al., 2011*), and its relevance to memory (*Burke et al., 2015*). To determine the specificity of PAC results to theta modulation of high-gamma, we computed MTL-precuneus PAC using a broader high-frequency spectrum (25–85 Hz). Permutation-based cluster-corrected t-tests comparing memory to rest during vertex stimulation sessions revealed a cluster of increased PAC encompassing 3 Hz modulation of 77–81 Hz gamma, and 4 Hz modulation of 59–81 Hz gamma (cluster p=0.018; t-sum = 58.02) (*Figure 4—figure supplement 1A*). During precuneus stimulation sessions, PAC was increased during memory relative to rest between 5 Hz theta and 61–81 Hz gamma (cluster p=0.031; t-sum = 75.93) (*Figure 4—figure supplement 1B*). Comparison of precuneus to vertex stimulation sessions revealed a cluster of reduced 4 Hz theta modulation of 63–79 Hz gamma (cluster p=0.05, t-sum = −22.24) (*Figure 4—figure supplement 1C*). These findings mirror the PAC results described above, and demonstrate the specificity of PAC to theta modulation of high-gamma, although note that the exact frequency range of high-gamma varies depending on the study and the method of electrophysiological recording used (cf. *Crone et al., 2011*; *Kaplan et al., 2014*).

### Exploratory analysis: causal role of precuneus

In a final exploratory analysis, we sought to clarify the causal role of the precuneus in complex memory retrieval. Specifically, we aimed to demonstrate that the precuneus plays a causal role in cross-regional oscillatory interactions important for memory, in line with our hypotheses. To do so, we

assessed whether cross-regional connectivity with the precuneus was associated with stimulation-induced changes in subjective memory. We exported average PAC values from the previously identified significant cluster during vertex stimulation sessions (*Figure 3C*) to obtain a putative measure of undisturbed MTL-precuneus communication. MTL-precuneus PAC during vertex stimulation sessions was significantly negatively correlated with stimulation-induced change in vividness (partial r = −0.554, p=0.008, CI [−0.77 −0.27]) when controlling for counterbalancing order, but not with change in effort rating (partial r = 0.314, p=0.154, CI [−0.09. 71]). MTL-TPJ and MTL-mPFC coupling were not associated with change in vividness or effort (all p's > 0.668). Thus, greater MTL-precuneus connectivity predicted greater stimulation-induced decreases in vividness. Change in phase coupling within the previously identified significant cluster was not significantly associated with change in vividness or effort rating (p's > 0.812). This finding demonstrates that MTL-precuneus functional connectivity is related to the observed effects of precuneus stimulation on subjective memory, pointing to a causal role for the precuneus in network-wide activity which is important for complex memory retrieval.

## Discussion

The present study revealed three main findings: (1) Recall of complex personal memories involves the precuneus as a critical cortical node; (2) Complex memory retrieval is supported by MTL-cortical communication mediated by theta phase coupling and theta-gamma PAC; (3) Inhibitory precuneus stimulation leads to network-level alteration of MTL-driven oscillatory coupling, both with the precuneus itself and with other posterior cortical structures. These novel findings demonstrate a causal role for posterior cortical regions in driving the mechanisms that support memory for complex events.

### Oscillatory mechanisms of complex memory retrieval

We first sought to characterize the oscillatory correlates of complex personal memory retrieval by examining activity at vertex stimulation sessions. We found theta power increases in an extensive network of regions including occipital lobes, precuneus, inferior and superior parietal lobes, retrosplenial cortex, MTL, mPFC, and cerebellum. While previous studies have mainly focused on theta oscillations in a constrained network of regions during autobiographical memory retrieval (*Foster et al., 2013*; *Fuentemilla et al., 2014*; *Steinvorth et al., 2010*), here we demonstrate that theta power across a widespread network of regions is associated with complex memory recollection. These findings converge with fMRI studies identifying a similar network of brain regions during autobiographical memory retrieval (for review, see *Svoboda et al., 2006*). They also suggest that theta oscillations may be a means of communication between regions in this widespread network, although note that increased power does not imply coupling. To examine communication between these regions we subsequently examined theta phase coupling and theta-gamma PAC.

Using a subject-specific seed in the right MTL, we identified a cluster in the occipital lobe that was theta phase synchronized with the seed during early memory elaboration. These findings indicate that the MTL communicates with posterior neocortical regions via theta phase coupling during memory recollection. Two previous studies have demonstrated network-level theta phase coupling as a correlate of autobiographical memory retrieval (*Foster et al., 2013*; *Fuentemilla et al., 2014*). *Fuentemilla et al. (2014)* identified theta phase coupling between an MTL seed and clusters in the precuneus and mPFC during AM recollection relative to a general semantic knowledge task. Differences in the specific cortical structures identified in our findings and theirs may be related to their use of continuous recordings of personal memories as they unfold (direct cues), whereas we used personalized cues that require indirect retrieval processes. Different baselines (semantic memory in Fuentemilla et al., and rest in the present study) are also likely contributors to the slight differences in the cortical structures identified across studies.

Theta phase coupling may provide a scaffold for interregional coordinated activity (*Fell and Axmacher, 2011*), whereas local gamma is considered an index of stimulus-specific information processing. We therefore also investigated whether local cortical gamma amplitude is modulated by the phase of MTL theta oscillations. We identified phase-amplitude coupling between MTL theta and precuneus gamma during memory elaboration compared to rest which was sensitive to vividness ratings, but did not find PAC between the MTL and mPFC or TPJ. Interestingly, MTL theta

modulated precuneus gamma but the reverse was not true, supporting the MTLs role in coordinating neocortical activity. A number of studies have identified theta-gamma coupling in human lab-based memory tasks (*Axmacher et al., 2010*; *Staudigl and Hanslmayr, 2013*; *Canolty et al., 2007*), but ours is the first to demonstrate this phenomenon in complex naturalistic memory retrieval. It is interesting that MTL theta was coupled with precuneus gamma but not with the other neocortical regions examined. Many previous studies have shown that medial prefrontal and lateral parietal regions are recruited during AM retrieval (for review, see *Svoboda et al., 2006*), and we similarly found robust theta power increases in these regions during recollection. However, our findings do not indicate that local gamma activity in these regions is coordinated by MTL theta, which is unexpected given the hypothesized early role of mPFC during retrieval of self-related memories and its connectivity with the MTL (*Hebscher and Gilboa, 2016*; *McCormick et al., 2018*). Further research is needed to elucidate the conditions under which communication between these nodes of the network become critical for retrieval. Our findings thus demonstrate that MTL-precuneus oscillatory coupling is important for memory recollection but leave open questions about communication across other nodes in the network. Together these results show that MTL and posterior neocortical regions interact via theta and gamma coupling during complex personal memory recollection, suggesting a specific means of information transfer between these distant regions.

## Effects of precuneus stimulation

Having demonstrated that theta and gamma oscillations mediate the communication between widespread regions during recollection, we next examined whether the precuneus is critically involved in this communication. We show that precuneus stimulation disrupts theta phase coupling between the MTL and occipital lobe, supporting a causal role for this region in theta phase coupling between regions beyond the site of stimulation. These findings are consistent with the idea that theta coordination of network activity mediates integrated representations required for the reinstatement of complex memories.

We further found that precuneus stimulation altered MTL-precuneus theta-gamma coupling during memory recollection relative to rest. Precuneus stimulation led to reduced 4 Hz theta modulation of gamma in a cluster that showed increased coupling during vertex stimulation sessions. Notably, MTL-precuneus communication within this cluster was initially correlated with vividness ratings, and stimulation-induced disruption of PAC within the cluster predicted disruption of vividness. These findings indicate that precuneus stimulation disrupted MTL-precuneus communication which was initially present during recollection, communication which was crucial for the subjective vividness of memories. Precuneus stimulation also increased 5 Hz theta modulation of gamma relative to vertex stimulation, an effect that corresponded with increased 5 Hz coupling during precuneus stimulation sessions. This stimulation-induced increase in 5 Hz coupling was not associated with change in subjective memory. One possible explanation for this stimulation-induced increase in coupling is that our cTBS protocol, consisting of high-frequency bursts applied at 5 Hz, entrained theta activity at 5 Hz. Indeed, previous studies have found that neurostimulation can synchronize oscillations to a specific frequency, although these studies used electrical stimulation (*Neuling et al., 2012*; *Zaehle et al., 2010*). Note, however, that this explanation is speculative and further research is needed to determine the feasibility of using cTBS to entrain neural oscillations.

We found tentative evidence to suggest that precuneus stimulation alters subjective memory vividness and ease of recall. Interestingly, we found that the order of counterbalancing significantly interacted with the effects of stimulation on memory vividness. This order effect prompted us to perform an exploratory between-subjects analysis comparing participant's first session, which revealed that inhibitory precuneus stimulation led to decreased memory vividness and more effortful recall compared to vertex stimulation. Comparison of participant's second session revealed no significant effect of stimulation, although the pattern of results was opposite to that of the first session. One possible explanation for these findings is that, due to the subjective nature of the rating scales, memories recalled in the second session were rated relative to memories recalled in the first session. We also found that counterbalancing order did not influence the effects of stimulation on phase coupling or PAC, further suggesting that this order effect may have been specifically related to the subjective nature of the rating scales.

Very few studies have used neurostimulation to examine the causal role of the precuneus in memory. One study found that high-frequency repetitive TMS to the precuneus modestly enhanced

episodic memory in patients with Alzheimer's disease (*Koch et al., 2018*), while an earlier study from the same group found that precuneus cTBS enhanced source memory retrieval (*Bonnì et al., 2015*). The only two TMS studies of AM to date both targeted the angular gyrus and reported reduced internal episodic details compared to vertex stimulation. *Thakral et al. (2017)* additionally found increased external semantic details, while *Bonnici et al. (2018)* found a reduction in the number of events recalled from a first-person perspective (*Thakral et al., 2017*). Our findings add to this limited literature by demonstrating a causal role for the precuneus and precuneus-MTL interactions in subjective aspects of complex personal memory retrieval.

Contrary to the proposed importance of MTL-precuneus communication in representing spatial information, precuneus stimulation did not affect the tendency to recall events from a first-person perspective. While we previously found that precuneus volume is positively associated with recalling autobiographical memories from a first-person perspective (*Hebscher et al., 2018*), the present study does not support a causal role for the precuneus in this function. One interpretation of these results is that the precuneus is involved in egocentric processing during complex memory retrieval, but not causally so, perhaps, due to its interactions with other posterior parietal regions like the angular gyrus. As described above, one recent study found that angular gyrus stimulation reduced the number of autobiographical memories experienced from a first-person perspective, while also reducing the number of internal details recalled (*Bonnici et al., 2018*). The authors interpret this result as implicating the angular gyrus in integrating memory features within an egocentric framework to enable the subjective experience of remembering. Other studies have implicated both the angular gyrus and precuneus in shifting visual perspectives during autobiographical memory (*Iriye and Jacques, 2018*; *St Jacques et al., 2017*). Thus, it may be the case that interactions between these regions are important for representing events from an egocentric perspective, with the angular gyrus playing more of a critical role than the precuneus. Future TMS studies are needed to clarify the nature of the precuneus' involvement in egocentric processing.

## Conclusions

We show that MTL and posterior neocortex interact via theta and gamma oscillatory activity during complex personal memory retrieval. Our results support the notion that theta phase coupling and theta-gamma phase-amplitude coupling mediate MTL-neocortical coordination during memory processes. We further show that precuneus stimulation alters oscillatory activity and subjective memory, demonstrating a causal role for this region. Our results indicate that continuous theta burst stimulation can be used to causally alter oscillatory activity, and that these effects are long-lasting. Together, these findings demonstrate the feasibility of using cTBS and MEG to study complex, naturalistic memory functions.

# Materials and methods

## Participants

Twenty-three healthy young participants (14 females, mean age = 26.3, range = 19–36) were tested on a within-subjects combined TMS-MEG paradigm. Sample size was determined by an a priori power analysis based on a previous TMS-EEG study of episodic memory (*Nilakantan et al., 2017*). Participants were recruited from the Rotman Research Institute's healthy volunteer pool. Participants had completed an average of 16.4 (range = 14–21) years of formal education, were all right-handed, native or fluent English speakers, had normal or corrected-to-normal vision, and were free from a history of neurological illness or injury, psychiatric condition, substance abuse, or serious medical conditions. Based on TMS safety guidelines (*Rossi et al., 2009*), participants were excluded if they had a history of losing consciousness (fainting), had a prior experience of a seizure, or had a diagnosis or family history of epilepsy. All participants provided informed consent prior to participating in the experiment in accordance with the Rotman Research Institute/Baycrest Hospital ethical guidelines.

## Procedure

Participants received cTBS to their left precuneus and to a control region (vertex) on separate days, at least 24 hr apart (mean = 5.4 days). Immediately following cTBS, participants completed the AM task inside the MEG scanner which was located nearby. All participants completed the MEG scan

within an average of 27.4 (SD = 3.8) min measured from the end of cTBS. Average time between the end of cTBS and start of MEG was 6.04 (SD = 1.5) min. Anatomical MRIs for each participant were collected in a separate session.

## Stimuli and task

At least 48 hr prior to the study, participants provided the names of familiar places, objects and people in an online interview. These items were used as cues because they are elements that commonly make up an event (*Addis et al., 2009*; *Burgess et al., 2001b*). Participants were instructed to name the first 20 items that came to mind and to limit items to those encountered within the past year.

Based on the online interview, 60 cue words were created for each participant, 20 per category. These were randomly divided between the two stimulation sessions so that each session included 30 cues (10 of each cue), and each session was broken down into three runs to be used in the MEG scanner. E-Prime 1.2 software was used to display the items and collect response data. Items were presented in a randomized order. Participants were instructed to use the words as cues to recall personal specific events that had occurred within approximately the last year, not including the past week. Specific events were defined as 'past events from a specific time and place for which you were personally involved.' Cue words were displayed for a maximum of 10 s and participants were instructed to retrieve a specific past event related to the cue as quickly as possible. Participants were asked to press a button on the response box corresponding to their right index finger as soon as a memory came to mind. Trials in which no memory was retrieved (unsuccessful trials) were discarded. The retrieval phase was terminated when a memory was retrieved, after which participants saw a slide asking 'What was the very first thing that came to mind', and had to choose one of the following four options: person, object, place, other. An elaboration phase followed in which participants were prompted to imagine the event in as much detail as possible for 8 s. Next, participants rated the memory on four scales aimed at measuring different phenomenological characteristics of the memory. They were given a maximum of 5 s per rating scale. Participants were asked to rate the effort required to bring the event to mind (1 = very easy, 6 = very effortful), feelings of re-experiencing the event (1 = not at all, 6 = completely), recall of setting (1 = not at all, 6 = distinctly), and perspective (1 = saw event through my own eyes, 6 = saw myself from an external perspective) (Scales adapted from *Addis et al., 2007*; *Arnold et al., 2011*). Participants were instructed to rate the experience of remembering and not the event itself. Response options on the screen appeared in square boxes representing the response box used in the MEG scanner, with each box representing one of the four buttons for each hand (excluding the thumb of each finger). Participants completed practice trials outside of the MEG to familiarize themselves with the task before moving on to the test trials. See *Figure 1* for a depiction of the autobiographical memory task.

## TMS procedure

Participants received cTBS to their left precuneus (MNI −14,–66, 56) and to a control region (vertex; MNI 0,–15, 74) on separate days. The order of these sessions was counterbalanced. Participants were blind to the type of stimulation they received (precuneus or vertex) and later interviews indicated they could not distinguish between the two. The precuneus was chosen as a target region based on this regions' involvement in AM and in representing spatial perspective during AM (*Freton et al., 2014*; *Hebscher et al., 2018*). We chose the left precuneus based on evidence showing that episodic/autobiographical memory is predominately associated with left-lateralized parietal activity (*Kim, 2011*; *Rugg and Vilberg, 2013*; *Shimamura et al., 2011*). The left precuneus target region was selected based on a custom meta-analysis of 13 studies with the keyword 'egocentric' using NeuroSynth (neurosynth.org). Within this map, the target region was selected by choosing the coordinates with the highest z-score that would be the most accessible with TMS (the most superficial area). Prior to stimulation, both stimulation site coordinates were warped from MNI to individual space and the stimulation site was chosen based on individual anatomy from whole-brain anatomical MRIs as the most superficial region that was closest to these coordinates.

At the beginning of the first stimulation session, resting motor threshold (RMT) was measured for each participant as the lowest intensity that produced motor evoked potentials (MEPs) above 50 μV in 5 out of 10 trials, recorded from the right first dorsal interosseous muscle. A Brainsight frameless stereotaxic neuronavigation system (Rogue Research, Montreal, Quebec, Canada) was used to

target the selected stimulation sites. Three anatomical landmarks located on the face were used to co-register the anatomical MRI to the participant's head. An infrared camera (Polaris Vicra, Northern Digital) recorded sensors attached to the participant and the TMS coil, allowing for real-time tracking of the TMS coil over the participant's MRI. A biphasic Super-Rapid Stimulator with a 70 mm air-cooled Figure 8 coil (Magstim Co., Whitland, Dyfed, UK) was used to deliver a modified continuous theta burst stimulation (cTBS) at 80% RMT, lasting for approximately 40 s. The cTBS protocol consisted of 600 pulses arranged into bursts delivered every 5 Hz (200 ms), with each burst containing three pulses delivered at 30 Hz. The coil was positioned perpendicular to the stimulation site. Although standard cTBS protocols are delivered at 50 Hz, we decided to lower the burst frequency to 30 Hz due to limitations of the coil circuitry, leading to overheating at high intensities. Reducing the frequency of stimulation allowed us to stimulate at a higher intensity than would be possible at 50 Hz. Similar protocols have previously been shown to induce stronger MEP suppression compared to 50 Hz cTBS at a reduced intensity (*Wu et al., 2012*).

## MEG acquisition

MEG was recorded in a magnetically shielded room at the Rotman Research Institute with a 151-channel whole-head system with first order axial gradiometers (CTF MEG, Coquitlam, BC, Canada) (VSM MedTech Inc), at a sampling rate of 625 Hz. Participants sat in an upright position and the behavioural task was projected onto a screen in front of them. For the first four participants, MEG data was recorded continuously during the 30 min behavioral task. We subsequently divided the behavioral task into three equal blocks approximately 10 min in length in order to reduce overall measures of head movement. To further minimize head movement, a towel was inserted to provide a tighter fit within the helmet. Head position was tracked at the beginning and end of each recording block by coils placed at three fiducial points on the head. Average head position across runs was used for source estimation and was co-registered with fiducial points marked on the anatomical MRI. After acquisition, continuous signals were divided into epochs corresponding to each trial.

## MEG analyses

### MEG source estimation

MEG data were analyzed in source space using the adaptive beamforming technique Synthetic Aperture Magnetometry (SAM) implemented in CTF software (CTF Systems Inc, Port Coquitlam, BC, Canada). SAM is a beamformer technique used to compute the time course of virtual channels on a grid of locations (voxels) across the brain. SAM constructs a spatial filter from the data covariance matrix and a lead field map derived from the MRI head model. These spatial filters can estimate the virtual signal of activity generated at the target location while minimizing signal power from all other locations. Virtual signals are estimated by multiplying the original sensor data with a set of beamforming weights to create a new, spatially filtered time series focused on an area of interest. For connectivity analyses, virtual signals were computed over the whole brain on a regular grid with a resolution of 10 mm$^3$. SAM can also be used to produce a map of differences in oscillatory power between two states. For whole-brain maps of power differences, a noise normalized 'pseudo-T' map (*Vrba and Robinson, 2001*) was produced, also with a resolution of 10 mm$^3$. Analysis in source space is preferred over raw MEG sensor data because the beamforming procedure attenuates artifacts generated outside the brain, such as eye movements. The beamforming procedure also compensates for differences across participants in head shape and position (*Olsen et al., 2013*). Furthermore, a number of studies have demonstrated that MEG source estimation can reliably detect hippocampal signals (For review, see *Pu et al., 2018*), allowing us to address the hypotheses outlined above.

### MEG task-induced changes in theta power

We first conducted a whole-brain analysis of task-induced changes in theta power. Data for each successful trial (trials in which a memory was retrieved within the allotted time) was epoched into early memory elaboration, defined by the screen instructing participants to 'imagine event in detail' (500 to 3000 ms). The first 500 ms of elaboration was not analyzed in order to minimize evoked effects occurring due to the onset of a visual stimuli ('imagine the event in detail' screen), and eye movements due to reading this text. We measured early elaboration to reduce between-subject variability present in longer, self-guided time windows. For MEG source analyses, covariance matrices were

calculated on data bandpass filtered in the theta band (3–7 Hz) within each time window for both precuneus and vertex stimulation sessions. These covariance matrices were then used to compute the pseudo-T SAM maps of power differences (described above). Using this technique, we created single subject maps quantifying the change in theta (3–7 Hz) power between the active time window (elaboration) and a baseline inter-trial rest time window of equal length. These maps were warped into MNI space for subsequent analyses.

## Phase coupling

We computed theta phase coupling to measure functional connectivity between regions. The Weighted Phase Lag Index (wPLI) of the source-estimated time series was calculated using a right MTL seed voxel. Lateralization of the seed voxel was determined by the location of maximum theta power increase between memory and rest within a bilateral MTL mask (as determined by SAM maps described above). Participant-specific seeds were chosen as the voxel with the greatest theta power increase between memory and rest within an anatomically defined right MTL mask. wPLI was calculated for memory elaboration and rest time windows separately, using beamforming weights computed on both time windows combined. wPLI is a modified version of Phase Lag Index (PLI), which measures the strength of the coupling between two time series, where values close to one indicate a consistent nonzero phase relationship, and values close to zero indicate randomly distributed phase differences (*Stam et al., 2007*). wPLI has been shown to be more robust than PLI and does not overestimate effects caused by volume conduction or noise (*Vinck et al., 2011*). We calculated whole-brain wPLI maps, measuring phase coupling between the seed voxel and every other voxel in the brain using the FieldTrip toolbox in Matlab. Summary whole-brain wPLI images for each participant were then subjected to statistical analyses in AFNI.

## Phase-amplitude coupling

We computed theta-gamma phase amplitude coupling, or multiplexing, to determine whether distributed regions communicate across frequency bands during AM retrieval. PAC was calculated between theta phase of a right MTL seed and gamma amplitude of left precuneus, right mPFC, and left TPJ ROIs. Lateralization of the seed and ROIs were chosen using the same method described above. Participant-specific MTL theta seeds were the same as used in phase coupling analyses. Cortical ROIs were determined based on the voxel with the greatest theta power increase between memory and rest within each anatomically defined mask. To obtain larger and more representative gamma seed regions, surrounding voxels within 10 mm of the peak voxel were also included. We computed PAC between the MTL seed and each neocortical voxel within ROIs separately, and subsequently averaged these values to obtain a representative estimate of PAC between the MTL and each neocortical seed region.

As a control analysis, we also performed PAC between the same neocortical ROIs and MTL seeds, but using the theta phase of the neocortical ROIs and the gamma amplitude of the MTL seed (the reverse computation). We hypothesized that MTL theta phase would modulate neocortical gamma amplitude based on the established role for the MTL in coordinating neocortical activity (*McClelland et al., 1995*; *Rolls, 2000*). We did not expect the reverse (neocortical theta phase modulates MTL gamma amplitude) to be true.

Time windows for elaboration were the same as used for power and phase coupling. Measures of theta phase $(f_p)$ and gamma amplitude $(f_a)$ were calculated using a fourth order, two-pass Butterworth filter, and then applying the Hilbert transform. Bandwidth filters used were [3 7] for theta and [65 85] for high-gamma. We chose to examine high-gamma based on its reported consistency across a variety of studies (*Crone et al., 2011*), and its relevance to memory (*Burke et al., 2015*). Consistent with a study demonstrating theta-gamma PAC during a spatial memory task using MEG, we defined our high-gamma frequency band as 65-85 Hz (*Kaplan et al., 2014*). Coupling between theta phase values $(\phi f_p)$ and the phase of gamma amplitude envelope $(\phi f_a)$ were calculated at each time point $(n)$ using a phase locking value modulation index (PLV-MI) (*Cohen, 2008*). Phase locking value (PLV) is similar to the wPLI calculated for theta phase coupling. PLV is computed as the vector length of phase differences over time from a signal (with length N), such that larger values reflect less phase difference variability between two signals.

$$MI = \left| \frac{1}{N} \sum_{n=1}^{N} e^{i\left(\phi f_p(n) - \phi f_a(n)\right)} \right|$$

We calculated PAC for each trial and averaged these to obtain PLV values for each amplitude and phase pair. The same procedure was repeated using surrogate data created by shuffling trial and phase information (200 surrogates), which we used to normalize PLV values. This resulted in surrogate-normalized comodulograms for each condition displaying the degree of PAC between each phase and amplitude value, which we then subjected to statistical analysis. PAC was computed using the FieldTrip toolbox in Matlab with code adapted from *Seymour et al. (2017)*.

## Statistical analyses

### Behavioral analyses

To test the hypothesis that precuneus stimulation would lead to differences in the quality of memory retrieval, we ran a series of ANOVAs using rating scales from the AM task for precuneus and vertex sessions as within-subjects factors and session order, (i.e. counterbalancing order: precuneus stimulation or vertex stimulation first) as the between-subjects factor, to account for potential effects of counterbalancing order on subjective memory (n = 23). Given the lack of experimental evidence on the effects of TMS on AM, we chose to investigate all rating scales. Rating scales for re-experiencing and setting were positively correlated for both sessions (r > 0.50), so the average of these scales was taken as a measure of vividness. We looked at the effect of precuneus stimulation on effort required to bring the event to mind (subsequently referred to as ease of recall), perspective (egocentric to allocentric), and vividness. We also performed post-hoc exploratory analyses on rating scales using each participant's first session, such that we compared participants with precuneus stimulation on day one (n = 12) to those with vertex stimulation one day one (n = 11) in a between-subjects design. We performed the same analysis for each participant's second session. Behavioral analyses were performed in RStudio.

### Theta power

To test the hypothesis that AM retrieval would be associated with increased theta activity, SAM maps for elaboration vs. rest within vertex stimulation sessions were analyzed using one-sample t-tests (n = 23). We used one-sample t-tests because each SAM map was itself a comparison between memory and rest. SAM maps for memory > rest were subsequently submitted to a paired t-test comparing precuneus to vertex stimulation. All statistics on whole-brain power changes were performed in AFNI.

### Phase coupling

To address the prediction that memory retrieval relative to rest would be associated with increased theta phase coupling, memory elaboration was compared to rest using paired-samples t-tests. To test the hypothesis that precuneus stimulation would alter phase coupling, wPLI maps were subjected to paired t-tests comparing precuneus to vertex stimulation for memory elaboration relative to rest (memory > rest) (n = 23). T-tests were performed in AFNI. Average wPLI values within significant clusters were subsequently exported for each participant and correlated with behavioral measures in RStudio.

### Phase-amplitude coupling

To test the prediction that MTL theta modulates neocortical gamma during AM retrieval, comodulograms for memory elaboration were compared to rest. To test the hypothesis that precuneus stimulation would alter PAC, comodulograms for elaboration (memory > rest) following precuneus stimulation were compared to those following vertex stimulation (n = 23). All comparisons were computed using non-parametric permutation-based statistics. First, a paired-samples t-test was performed between conditions (memory vs. rest; precuneus vs. vertex stimulation). Monte Carlo estimates of the significance probabilities were calculated by randomly permuting comparisons over 1000 iterations, using a cutoff threshold of p<0.05. All values exceeding this threshold were grouped into clusters, and the maximum t-value within each cluster was compared against a null distribution

obtained by shuffling the data 1000 times and calculating the largest cluster-level t-value for each permutation. A threshold of p<0.05 was used for the permutation-based cluster-level comparison. Test statistics for this analysis reflect the sum of t-values for each value within a cluster (t-sum). Statistical analyses were performed using the FieldTrip toolbox in Matlab with code adapted from *Seymour et al. (2017)*. Average PAC values within significant clusters were subsequently exported for each participant and correlated with behavioural measures in RStudio.

### Cluster corrections

SAM and wPLI maps were thresholded at p<0.005 using a minimum cluster size to keep the family-wise error rate at p<0.05. Cluster sizes were determined based on the spatial smoothness of the data using Monte Carlo simulations from the 3dClustSim tool in AFNI using FWHM values calculated based on noise maps produced by conducting a voxel-wise t-test on single-subject maps comparing the pre-trial rest period between two randomly selected cue-types (locations and people). As different maps have different smoothness values, we obtained cluster sizes separately for each map. Cluster thresholds were 78 voxels (FWHM 21.8) for SAM theta maps and 24 voxels (FWHM 11.6) for wPLI maps.

## Acknowledgements

We thank Christine Ibrahim, Brahm Sanger, and Kyle Nealy for their assistance in testing participants, and John Griffiths for help with MEG analyses.

## Additional information

### Funding

| Funder | Grant reference number | Author |
| --- | --- | --- |
| Natural Sciences and Engineering Research Council of Canada | Discovery Grant 378291 | Asaf Gilboa |
| Natural Sciences and Engineering Research Council of Canada | Postgraduate Scholarship-Doctoral | Melissa Hebscher |

The funders had no role in study design, data collection and interpretation, or the decision to submit the work for publication.

### Author contributions

Melissa Hebscher, Conceptualization, Data curation, Formal analysis, Funding acquisition, Validation, Investigation, Visualization, Methodology, Writing—original draft, Project administration, Writing—review and editing; Jed A Meltzer, Resources, Formal analysis, Supervision, Validation, Methodology, Project administration, Writing—review and editing; Asaf Gilboa, Conceptualization, Resources, Formal analysis, Supervision, Funding acquisition, Validation, Methodology, Project administration, Writing—review and editing

### Author ORCIDs

Melissa Hebscher ⓘ http://orcid.org/0000-0002-1863-5464

### Ethics

Human subjects: The study was approved by the Rotman Research Institute/Baycrest Hospital ethics committee (REB #16-33). All participants provided informed consent prior to participating in the experiment.

### Decision letter and Author response

Decision letter https://doi.org/10.7554/eLife.43114.017
Author response https://doi.org/10.7554/eLife.43114.018

## Additional files

### Supplementary files
• Transparent reporting form
DOI: https://doi.org/10.7554/eLife.43114.013

### Data availability

All data generated during this study are included in the manuscript and supporting files. Source data files have been provided for Figures 2, 3C, and 4C.

The following dataset was generated:

| Author(s) | Year | Dataset title | Dataset URL | Database and Identifier |
|---|---|---|---|---|
| Hebscher M, Meltzer JA, Gilboa A | 2018 | Data from: A causal role for the precuneus in network-wide theta and gamma oscillatory activity during complex memory retrieval | https://dx.doi.org/10.5061/dryad.gf4f363 | Dryad Digital Repository, 10.5061/dryad.gf4f363 |

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
