## [Decision Letter]

Thank you for submitting your article "A causal role for the precuneus in network-wide theta and gamma oscillatory activity during complex memory retrieval" for consideration by *eLife*. Your article has been reviewed by Laura Colgin as the Senior Editor, a Reviewing Editor, and three reviewers. The following individuals involved in review of your submission have agreed to reveal their identity: Joel Voss (Reviewer #1); Michael X Cohen (Reviewer #2).

The reviewers have discussed the reviews with one another and the Reviewing Editor has drafted this decision to help you prepare a revised submission.

Editors’ Note: Joel Voss not further included in the evaluation after declaring a competing interest.

Summary:

This study tests the causal role of different oscillatory frequencies in cortical-medial temporal lobe (MTL) networks for autobiographical memory. An innovative combination of magnetoencephalography and continuous theta burst stimulation is used to test these network-wide effects. The results indicate that theta-gamma phase-amplitude coupling is associated with memory retrieval and the precuneus may play a causal role in establishing these dynamics.

Essential revisions:

The reviewers were in agreement that this study has yielded a potentially impactful contribution to the literature on autobiographical memory. As one reviewer noted, the precuneus has a long association with autobiographical memory but the unique approach in the present paper can provide new insights into its causal role within a dynamic network. Nevertheless, the reviews and discussion highlighted a number of essential revisions that would need to be addressed for the results to be fulling convincing.

1) The causal relationship of the oscillatory activity to the behavioral components of autobiographical memory (AM) should be better established. Much of this stems from the general point that though stimulation was observed to have affected theta-gamma coupling and vividness in separate analyses, the relationship of these to each other was not established. In other words, there is not evidence that the impact of stimulation on a behavioral feature like vividness was caused by its impact on PAC. This general problem underlies two related issues:

First, it is yet unclear what it is about precuneus stimulation that led to diminished vividness of AM. One possibility is due to actual impairment of the memory retrieval process. Specifically, perturbation of MTL-cortical communication may degrade the efficacy of memory retrieval process, resulting in subjective perception of poorer vividness. Alternatively, precuneus stimulation may be affecting local activities in the precuneus that itself is associated with subjective experience/evaluation of memory (such as confidence and vividness) without affecting the efficacy of memory retrieval process itself.

Second, it is unclear whether stimulation effects on vividness and MTL-precuneus PAC are parallel in the analyses. Vividness of AM was affected by precuneus stimulation depending on the counterbalancing order of stimulation (i.e., subsection “Behavioural results”, significant interaction between session order and stimulation type), and there was a main effect of stimulation type on vividness when session order was accounted for (subsection “Behavioural results”). Given the significant influence of counterbalancing order on the stimulation effects on vividness, simply contrasting PAC between precuneus and vertex stimulation sessions without accounting for the counterbalancing order hinders establishing a clear causal link between the stimulation-driven changes in vividness and PAC measures.

Analysis of correlations between the changes of AM and PAC measures induced by stimulation or performing some form of mediation analysis that would establish a link between stimulation, PAC, and the behavioral outcomes would help addressing the general concern, as well as these more specific ones.

2) The localization and impact of stimulation effects to precuneus is unclear. The stimulation of precuneus with cTMS does not ensure that local inhibition of precuneus is the cause of any observed effects. Because the effects were distributed, the precuneus may have served as nothing more than a conduit. If so, its causality would take on a very different meaning than that favored in the paper. Related to the above point, it would be very helpful to conduct analyses to determine if any of the observed changes (memory vividness, distributed oscillatory activity/synchrony) are related to the precuneus. One could determine whether an area's connectivity with the precuneus is a valuable predictor, or whether the local effect on precuneus activity predicts the downstream effect on other network locations. The latter would be required to infer a causal influence of the precuneus. At a minimum, it seems an important caveat without some additional evidence on localization.

3) The range for the gamma band should be broader. It is important to determine whether there is spectral specificity to a gamma band, or whether the results span a broader high-frequency spectrum that includes gamma.

4) The 4Hz to 5Hz shift is difficult to understand functionally or physiologically and requires further attention. The manuscript notes that this could signal entrainment, or perhaps compensation. However, a compensation-related explanation is not entirely obvious (why would compensation push coupling to the specific frequency that was applied?) There was some concern that it was produced by the double-subtraction. The authors need to break down the analysis plotted in Figure 4 further to show PAC in each of the TMS conditions separately. Was the shift correlated with effects on vividness?

---

## [Author Response]

Essential revisions:1) The causal relationship of the oscillatory activity to the behavioral components of autobiographical memory (AM) should be better established. Much of this stems from the general point that though stimulation was observed to have affected theta-gamma coupling and vividness in separate analyses, the relationship of these to each other was not established. In other words, there is not evidence that the impact of stimulation on a behavioral feature like vividness was caused by its impact on PAC. This general problem underlies two related issues:First, it is yet unclear what it is about precuneus stimulation that led to diminished vividness of AM. […]Second, it is unclear whether stimulation effects on vividness and MTL-precuneus PAC are parallel in the analyses. […]Analysis of correlations between the changes of AM and PAC measures induced by stimulation or performing some form of mediation analysis that would establish a link between stimulation, PAC, and the behavioral outcomes would help addressing the general concern, as well as these more specific ones.

Thank you for these comments. We agree with the reviewers that it is important to establish a link between the reported effects of stimulation on coupling measures and behavioural outcomes. To address the first point, we have added a series of analyses correlating stimulation-induced changes in coupling measures with changes in the behavioural measures affected by stimulation (vividness and effort). We calculated partial correlations between change in subjective memory and change in phase coupling (subsection “Effects of precuneus stimulation on phase coupling”), and PAC (subsection “Effects of precuneus stimulation on PAC”), holding counterbalancing order constant. Accounting for counterbalancing order in these partial correlations also helps address the second point. The results from these analyses show that stimulation-induced change in vividness is associated with stimulation-induced change in PAC, but not phase coupling, when holding counterbalancing order constant. These results also hold when counterbalancing order is not held constant (r =.450, p =.031). These findings establish a link between precuneus stimulation’s effects on PAC and vividness, and show that counterbalancing order does not significantly affect this relationship. Thus, perturbation of MTL-precuneus communication, specifically via phase-amplitude coupling, appears to reduce subjective memory vividness.

To further address the second point regarding the effect of counterbalancing order on PAC, we added control analyses to the text (subsection “Control analysis: Counterbalancing order”). These analyses examine the effect of counterbalancing order on the observed stimulation-induced changes in coupling, and show that changes in coupling measures remain when counterbalancing order is accounted for. We have also included a discussion of why counterbalancing order may have affected subjective ratings but not coupling measures, proposing order effects are more likely to reflect psychological factors such as biases and strategies in participants subjective rating scales than physiological ones. (subsection “Effects of precuneus stimulation”).

2) The localization and impact of stimulation effects to precuneus is unclear. The stimulation of precuneus with cTMS does not ensure that local inhibition of precuneus is the cause of any observed effects. Because the effects were distributed, the precuneus may have served as nothing more than a conduit. If so, its causality would take on a very different meaning than that favored in the paper. Related to the above point, it would be very helpful to conduct analyses to determine if any of the observed changes (memory vividness, distributed oscillatory activity/synchrony) are related to the precuneus. One could determine whether an area's connectivity with the precuneus is a valuable predictor, or whether the local effect on precuneus activity predicts the downstream effect on other network locations. The latter would be required to infer a causal influence of the precuneus. At a minimum, it seems an important caveat without some additional evidence on localization.

We agree with the reviewers that it is important to clarify the localization of stimulation effects. We want to emphasize that we believe the precuneus plays a causal role in oscillatory communication between regions during memory retrieval, and apologize for not making our position clearer in the original manuscript. Coordinated communication between regions is thought to be crucial for complex personal memory retrieval. This was emphasized by our initial goals and hypotheses outlined in the Introduction: “In the present study, we used cTBS to directly suppress neural activity in the precuneus in order to observe system-level changes in activity, specifically in the MTL and other structures that comprise the autobiographical memory network.”, and “We therefore predicted that precuneus stimulation would affect neural activity both within the precuneus and in regions functionally connected to the precuneus.”.

To further explore this, we have added an exploratory analysis correlating MTL-precuneus connectivity during vertex stimulation sessions with the effects of stimulation on vividness, controlling for counterbalancing order (subsection “Exploratory analysis: Causal role of precuneus”). The results from this analysis show that individual differences in MTL-precuneus connectivity predict the behavioural effect, such that those displaying greater connectivity under normal conditions also experience larger stimulation-induced reduction in vividness. This further supports our view that the precuneus plays a causal role in oscillatory communication between regions, which is important for memory retrieval.

3) The range for the gamma band should be broader. It is important to determine whether there is spectral specificity to a gamma band, or whether the results span a broader high-frequency spectrum that includes gamma.

To demonstrate that these effects are specific to high-gamma, we have added a control analysis using a broader high frequency band of 25-85 Hz (subsection “Control analysis: Gamma band specificity”), and included these results as a figure supplement to Figure 4 (Figure 4—figure supplement 1). The results from this analysis show significant clusters at high-gamma, extending slightly lower than our 65 Hz cut-off, mirroring the original findings. We also note that definitions of high-gamma vary depending on study and method of electrophysiological recording used.

4) The 4Hz to 5Hz shift is difficult to understand functionally or physiologically and requires further attention. The manuscript notes that this could signal entrainment, or perhaps compensation. However, a compensation-related explanation is not entirely obvious (why would compensation push coupling to the specific frequency that was applied?) There was some concern that it was produced by the double-subtraction. The authors need to break down the analysis plotted in Figure 4 further to show PAC in each of the TMS conditions separately. Was the shift correlated with effects on vividness?

We agree that this pattern of results is difficult to understand and apologize for not providing a clear explanation of our interpretation of them. As per the reviewer’s suggestion, we have included a breakdown of PAC in each stimulation session separately in Figure 4B, and have added statistics for PAC during precuneus stimulation sessions to the text (subsection “Effects of precuneus stimulation on PAC”). This breakdown helps clarify the pattern of results by showing that the stimulation-induced decrease at 4 Hz represents a disruption of 4 Hz PAC present during vertex sessions, while the stimulation-induced increase at 5 Hz is related to an increase in 5 Hz PAC during precuneus sessions.

Interestingly, as reported in response to point 1, only the change in 4 Hz PAC was correlated with change in vividness, suggesting that disrupting the ‘natural’ rhythms in this circuit alters memory. On the other hand, increased 5 Hz PAC during precuneus stimulation sessions may reflect entrainment to the stimulation protocol, but this does not appear to be behaviourally relevant to our task. We have included a discussion of this in subsection “cts of precuneus stimulation”. Finally, we have also removed the compensatory explanation as we agree it there is no clear prediction for it.